# Assessing Consumer Preferences for New Red-Pulp Kiwifruit: Application of a Choice Experiment between Different Countries

**DOI:** 10.3390/foods12152865

**Published:** 2023-07-27

**Authors:** Ivana Bassi, Giovanni Mian, Stefania Troiano, Enrico Gori, Luca Iseppi

**Affiliations:** 1Department of Agricultural, Food, Environmental and Animal Sciences, University of Udine, Via delle Scienze 206, 33100 Udine, Italy; ivana.bassi@uniud.it (I.B.); luca.iseppi@uniud.it (L.I.); 2Department of Economics and Statistics, University of Udine, via Tomadini, 30/a, 33100 Udine, Italy; stefania.troiano@uniud.it (S.T.); enrico.gori@uniud.it (E.G.)

**Keywords:** *Actinidia*, consumers, fruits, marketing, selection

## Abstract

The central objectives of this paper are to enhance the understanding of how consumers in developed economies value credence attributes and to understand their preferences for red-pulp kiwifruit. To achieve this, we utilised the choice experiment method through surveys conducted in Italy, Spain, France, and Germany, targeting kiwifruit consumers through specific questionnaires. Regarding red kiwifruit, a significant percentage of those who are already familiar with them either purchase or intend to purchase them. What is equally interesting is the high percentage of those who declared themselves to be undecided about making a purchase. Specific marketing actions can be directed towards the following two categories: converting the intention to purchase into an actual purchase and shifting the current inclination towards an intention or act of purchase, for example, by improving the knowledge about this relatively unknown fruit. This paper contributes to the market chain by assessing consumers’ choice and willingness to pay for red kiwifruit, while also comparing developed economy markets.

## 1. Introduction

Kiwifruit (*Actinidia* spp.), native to China’s Sichuan province (Yangtze River Valley), has become a popular fruit among consumers worldwide, and the largest collection of *Actinidia* germplasm resources is still preserved there [1]. Presently, kiwifruit is extensively cultivated across the globe, both in the Northern and Southern Hemispheres. According to statistics from the Food and Agriculture Organization of the United Nations (FAO), the global kiwifruit cultivation area and output have seen growth rates of 71.25% and 55.58%, respectively, over the past 10 years [2]. Although it faces significant phytoiatric problems resulting in reduced production [3], the demand for the fruit is still increasing [4], indicating a growing consumer interest.

China has the largest kiwifruit cultivation, estimated at approximately 1,800,000 tons, making it the leading producer worldwide. In the European Union countries, around 0.5 million tons are produced, primarily in Italy, France, Spain, Greece, and Portugal. Italy stands as the second-largest producer of kiwifruit globally, yielding nearly 496,000 tons, followed by New Zealand with 414,000 tons [5]. Other important producers are Chile and Greece.

Since 2007, the cultivated hectares of kiwifruit have been steadily increasing in Italy and worldwide [6]. Kiwifruit holds a significant position in the fresh fruit trade, boasting essential characteristics, with high levels of vitamin C (causing it to be recognised as a superfood) [7,8], which aligns with consumer demands and specific food product requirements. Additionally, it offers a medium to long shelf life, enabling the fruit to be sold at distant locations from cultivation sites, in varying ripening stages, and with extended transport possibilities [9]. As a result, it ranks among the top twenty internationally traded fruit products [4]. Moreover, two-thirds of the global production is allocated for export, and in Italy, after grapevine cultivation, which is the main perennial crop, kiwifruit is the most remunerative crop, and it is especially concentrated in the Friuli Venezia Giulia region [10,11]. However, despite the consistent increase in commercial flows, the market remains highly concentrated, with Italy, New Zealand, and Chile managing 75% of the production [12].

*Actinidia* spp. comprises numerous species (approximately 65) [13], all of which are edible. However, for over 30 years, the kiwifruit market has primarily focused on the cultivation of a single variety, Hayward (*Actinidia deliciosa*), which has green pulp [3,14]. Fortunately, there has been an increase in the number of varieties due to genetic breeding projects [15]. Broadly speaking, the varieties can be categorised into green pulp, yellow pulp, and two-coloured pulp [7]. While the green- and yellow-pulp varieties are well known to consumers, a new type, namely the red-pulp kiwifruit, is emerging in the market. Among the red-pulp varieties (*Actinidia chinensis*), the Hongyang cultivar is the most widely cultivated and traded. Research has indicated that this red-pulp kiwifruit cultivar is particularly suitable for fresh consumption due to its high sugar-to-acid ratio (10.39), which is an important factor for consumers [16]. Another variety that has garnered attention is Qihong cv. Both varieties were developed through breeding projects, resulting from a cross between Cuiyu (green pulp) and Kuimi (yellow pulp). Although red-pulp kiwifruits are not extensively traded or cultivated currently, they are gaining prominence within the kiwifruit industry due to their novelty and appeal to consumers.

When it comes to consumers’ preferences for purchasing food products, most consumers prioritise fruits with a perfect appearance, undamaged packaging, and a long shelf life [17]. Similarly, Helmert et al. (2017) [18] discovered that consumers consider quality, safety, and various other factors when selecting foods and fruits. Symmank et al. (2018) [19] provided an example showing that the appearance of bananas influences purchase intentions among German consumers, who also value the shelf life of bananas. Price is another factor that consumers take into account when making product choices. Helmert et al. (2017) [18] noted in their research that price labels can influence the attention, cognitive processing, and purchase intentions of European consumers regarding food products. Since the development of the random utility theory by Thurstone (1927) [20], attempts have been made to estimate models for individual decision makers [21]. In this regard, the choice experiment method (CEM) has emerged as an important tool for evaluating the value of fruits. The CEM, initially proposed by Louviere and Hensher [22,23], is based on the theoretical framework of the random utility theory [19]. It allows for the evaluation of goods or services based on their relevant characteristics, referred to as “attributes.” The CEM has been extensively used in non-market value evaluations, including studies focused on assessing the value of fruit.

Choice experiment studies that elicit values for quality attributes provide valuable insights for fruit agribusiness managers in various ways. Firstly, they help orchard managers determine if novel cultivars possess attributes that are highly valued by consumers. Secondly, warehouse and retail store managers can use the findings to apply appropriate techniques that enhance the fruit quality characteristics that are the most appealing to consumers. Lastly, marketing managers can utilise research-based information on consumers’ preferences to shape their strategies. For example, Tait et al. (2016) [24] employed a CEM approach to investigate the willingness to pay for mutton products with an environmental label certification among British consumers.

Based on these foundations, the main objective of this paper is to enhance our understanding of how consumers in developed economies value credence attributes and, more specifically, their preferences for red-pulp kiwifruit. To achieve this, we employ the choice experiment method (CEM) through surveys conducted among kiwifruit consumers in Italy, Spain, France, and Germany, employing specific questionnaires, which has been previously established as a suitable strategy and method [25,26,27,28].

## 2. Materials and Methods

To identify respondents’ preferences and determine their “willingness to pay” for specific characteristics of red kiwi, the “Choice Experiments” (CEM) method is employed [29,30]. This methodology combines random utility theory, consumer theory, experimental design theory, and econometric analysis [25]. The CEM allows for an investigation of respondents’ willingness to pay for specific attributes and provides insights into the trade-offs between different evaluated attributes. It is widely utilised in the agri-food sector, as evident from the extensive amount of the literature on this subject. Previous studies in the fruit domain have primarily focused on attributes such as appearance, date, and packaging, as well as consumer preferences for different food types.

In a choice experiment, several sets or choice groups are constructed. Each choice set presents a purchasing scenario to a respondent, featuring a variety of alternatives to choose from. Choice sets typically consist of two or more goods or services, each characterised by different combinations of attributes, including a monetary component (price or cost). With the CEM, the evaluation of a good or service is based on its most relevant characteristics, referred to as “attributes.” This method operates on the premise that goods or services can be described by their attributes, and an individual’s assessment depends on the intensity levels associated with these attributes in the offered product or service. For each attribute, which is deemed relevant and selected through stakeholder discussions, a defined set of levels is established [31].

For the specific case study being investigated, the choice experiment methodology (CEM) was employed. The process began by identifying the relevant attributes through focus group discussions with stakeholders. Subsequently, the levels of these attributes were defined to facilitate a comprehensive and thoughtful examination of the red-pulp kiwifruit. By combining different levels, an appropriate experimental design was formulated, and choice sets were constructed to capture respondents’ preferences. Given the numerous combinations of attributes and levels, a fractional factorial orthogonal design was generated using SPSS^®^ software [32]. A total of six choice sets were presented to respondents, each consisting of three alternatives representing different “types” of red-pulp kiwis characterised by varying levels of the identified attributes, including price.

CEMs involve survey questionnaires in which respondents are presented with the aforementioned choice sets, with each alternative described by the attributes and their respective level [29]. Respondents were asked to indicate which alternative they would choose if they were making an actual purchase. An opt-out alternative (“None of the proposed kiwis”) was included to account for the respondents’ freedom of choice in real market situations, where they may also decide not to purchase any kiwis at all. It was assumed that respondents, guided by the axioms of rationality and monotonicity of preferences, selected the alternative they perceived to be the best, providing them with the highest utility or satisfaction. In this study, five attributes of red kiwis were considered based on the findings of stakeholder focus group discussions. The attributes and their corresponding levels are presented in Table 1.

The “Area of origin” attribute had the following three levels: produced in Friuli Venezia Giulia (FVG), a region in Northeast Italy bordering Austria and Slovenia; produced in other Italian regions; and produced in other (foreign) countries. Within the Italian context, only the FVG region was chosen because it holds significant importance in terms of kiwifruit production and consumption, making it the most valuable and reliable region for the study. Additionally, red-pulp kiwis are not yet widely available in all Italian regions, as they are a new food product in both FVG and the Italian market. Hence, only this specific Italian region was taken into account, considering its recent emergence as the origin of red-pulp kiwis.

The “Brand” attribute considered the following two options: a leading international company and a local business.

The “Production method” attribute presented respondents with a choice between conventional methods using intensive techniques, and organic production adhering to specific standards and ensuring complete control over the production process.

The “Certification” attribute included the following two levels: presence or absence. During the focus group discussions, stakeholders emphasised the potential importance of certification. Therefore, an attribute related to either product quality or environmental sustainability certification was incorporated and described.

The “Price” attribute encompassed three levels, ranging from EUR 4.00 to EUR 8.00 per kilogram, as suggested by stakeholders.

Indeed, Table 2 presents the questionnaire used in this research. The questionnaire aims to collect information regarding the consumption of kiwifruit. It was structured into the following three sections: The first section aims to gather general characteristics of the respondents. The second section focuses on respondents’ attitudes towards fresh fruit, specifically kiwifruit. The third section explores preferences towards different pulp colours.

Before conducting the choice experiment method (CEM), the questionnaire included a brief explanation of the CEM itself. Respondents were instructed to imagine themselves in a situation where they had to evaluate the purchase of red-pulp kiwis based on their preferences for the attributes mentioned earlier, taking into account their budget. The attributes and levels were explained to the respondents before they proceeded with the choices, ensuring that they were familiar with the meanings associated with these characteristics. The various alternatives of red-pulp kiwifruit were presented to respondents using three identical, basic, and colourless designs. Each respondent was provided with six choice sets, consisting of three alternatives along with the option of “none of the proposed kiwifruit.” Each choice set required respondents to choose from three different kiwis, defined according to the considered attributes, and the” opt-out” alternative to give the respondents the freedom of choice that they have in real market situations, where they can also decide not to purchase any kiwis at all. They were asked to consider the choice sets as separate situations and answer each choice task. The data were collected by Dynata, an international provider of first-party data (data that are directly collected by the company) supplied by consumers and other individuals through telephone interviews.

An example of a choice set is depicted in Figure 1.

A total of 7212 observations were collected, comprising six choice sets per questionnaire and 1202 respondents. The CEM data were analysed using Nlogit 6.0^®^ software (Australia. 6/16 Carr Street, Waverton, NSW 2060, Australia). Initially, a multinomial logit model (MNL) was applied. Subsequently, the dataset was further analysed using a latent class model (LCM) to account for preference heterogeneity and improve the description of consumers’ preferences. The LCM allows for the possibility of segmenting purchase preferences into different consumer groups. This flexible method, implemented through a single questionnaire, enables the evaluation of respondent behaviour in various hypothetical scenarios, facilitating the exploration of preferences for different characteristics of the evaluated goods. By examining the choices made by respondents, it becomes possible to identify the underlying utility function and understand which characteristics and levels consumers believe enhance or diminish their satisfaction.

The utility function considered for both models is as follows:U(xi) = ASC + b1 × FVGi + b2 × LOCALEi + b3 × CERTi + b4 × BIOi + b5 × PRICE(1)
where FVG is a dummy variable for the Friuli Venezia Giulia region of origin, LOCALE is a dummy variable for a local company’s brand, CERT is a dummy variable for the presence of a quality or sustainability certification for red-pulp kiwifruit, BIO is a dummy variable for organic cultivation, and PRICE represents the price of red-pulp kiwifruit expressed in EUR/kg. ASC is a dummy variable for “none of the proposed kiwifruit”, included to take into account the utility derived from non-choice or other variables not included in the analysis.

## 3. Results

### 3.1. Descriptive Analysis of Sample

The survey included a sample of 1202 respondents equally distributed across Italy, Spain, France, and Germany (300 units each for Italy and Germany, and 301 units each for France and Spain). Descriptive statistics techniques were employed to analyse the respondents’ answers. The following tables present the breakdown of the Italian sample by region of origin and provide an overview of the entire sample based on gender, age groups, level of education, monthly income class, and occupation.

Regarding fresh fruit consumption, the majority of respondents stated that they purchase fresh fruit regularly, both for themselves and for others. More than half of the respondents reported consuming fresh fruit almost every day, with 89.1% consuming it at least 2–3 times per week. Only a small percentage (0.9%) indicated that they rarely consume fresh fruit. In terms of purchasing frequency, approximately half of the sample (49.3%) reported buying fresh fruit at least once a week, with the majority of purchases occurring on a weekly basis (25%).

When it comes to kiwifruit consumption, a significant portion of respondents (22.1%) reported eating them 2–3 times a week, while 17.5% consumed them at least once a week, and 22% consumed them at least once a month. Among the respondents who did not consume kiwifruits (7.7% of the sample), the main reasons cited were an unpleasant taste, specifically the sourness, and a preference for other fruits. Cost and the availability of kiwifruits through commercial channels were not significant obstacles to consumption.

In terms of knowledge about the different types of kiwifruits, the majority of the sample (72.3%) indicated an awareness of yellow-pulp kiwifruit, with variations observed between countries. Yellow-pulp kiwifruits were more widely known in Spain and France compared to Germany and Italy. Approximately 54% of the sample had knowledge of organic kiwifruit, while the awareness of red-pulp kiwifruits was relatively low at 12.2%. Italy showed the highest percentage of awareness for red-pulp kiwifruits, with 17.7% of respondents indicating knowledge of them. Among those who were aware of yellow-pulp kiwifruits, 86.8% reported regularly purchasing them. For red-pulp kiwifruits, 64.6% of respondents purchased them, with Italy having the highest percentage (71.7%). Finally, regarding organic kiwifruits, 89.1% of respondents stated that they purchased this type.

The whole sampling data are depicted in Appendix A.

### 3.2. Preferences for Red-Pulp Kiwifruit

The coefficients (b1 to b5) in the utility function can be interpreted as the marginal utilities of each attribute. The values presented in Table 3 represent the average shared coefficients for all respondents.

Before discussing the results of the latent class analysis, it is important to understand the findings of the base model, which highlight the attributes that are the most relevant for consumer choice on average and are driving industry strategies.

All coefficients obtained from the multinomial logit base model are statistically significant. The signs of the coefficients align with the expectation that the price coefficient has a negative sign, indicating that as the price of kiwi increases, the respondents demand a lower quantity, which is in line with consumer rationality.

The results demonstrate that the most important attribute for respondents when choosing red-pulp kiwi is the brand of a local company. Respondents show a particular appreciation for this feature in their purchasing decisions. Following in importance are the organic cultivation of red-pulp kiwis, and to a lesser extent, the presence of a quality or sustainability certification.

The attribute with the lowest marginal utility is the origin region of Friuli Venezia Giulia, which is linked to the presence of respondents who are not residents of that region. The “ASC” variable is statistically significant and negative, indicating that the respondents derive greater utility from choosing one of the proposed kiwis rather than not making a choice.

Based on the base model results, it is interesting to note that the presence of a local brand, as opposed to an international one, is an important characteristic for respondents when selecting red-pulp kiwi. Additionally, the respondents show significant interest in organic cultivation, which is possibly driven by the perceived positive externalities that are associated with this production method, which benefit the entire community.

The base model assumes no heterogeneity in the choices among the respondents, which is a limitation that can be addressed by developing alternative models. To address this, the estimation results of the latent class logit model are presented below.

Based on specific indicator values, a model was identified that revealed the existence of three latent classes, with probabilities of respondents belonging to each class at 19%, 45%, and 36%, respectively. The model exhibits good interpretative ability, as indicated by the obtained MacFadden Pseudo R-squared value of 39%, which is considered quite good given the nature of the models being considered.

The model successfully identifies two significantly sized groups (Classes 2 and 3) with probabilities of belonging at 45% and 36%, respectively, along with a smaller class (Class 1) with a 19% probability of belonging.

### 3.3. Analysis of the CLASSES Considered in This Study

By examining the coefficients of the variables, it is evident that there is heterogeneity among the different classes in terms of their preferences for the attributes used in the CEM.

Individuals belonging to CLASS 1 (with a 19% probability of belonging) place importance on the presence of a quality or environmental sustainability certification. They are willing to pay a premium price of EUR 1.10 per kilogram for this characteristic. They also value kiwis from the Friuli Venezia Giulia region, with a willingness to pay EUR 0.98 per kilogram, as well as organic cultivation, with a willingness to pay EUR 0.91 per kilogram. These individuals are particularly price sensitive. Female respondents are more likely to be part of this class, while individuals in the younger age group (18–24 years) and those who consume fresh fruit daily or almost daily are less likely to belong to this class. This group can be described as “supporters of local sustainability.”

CLASS 2 (with a 45% probability of belonging) prefers the presence of a local enterprise brand over an international one. They are willing to pay EUR 5.30 per kilogram for this characteristic of kiwis. This class is more likely to consist of young female respondents between the ages of 18 and 34, residing in Italy. They also appreciate organic production methods, with a willingness to pay EUR 3.44 per kilogram for this attribute, and are sensitive to price. This group can be identified as “confident in local brands.”

CLASS 3 (with a 36% probability of belonging) comprises consumers who prioritise the presence of a quality or sustainability certification for kiwis. This characteristic triggers a willingness to pay an additional EUR 1.22 per kilogram. They also appreciate organic kiwis, as indicated by their willingness to pay a premium price of EUR 0.42 per kilogram, and are attentive to price. This group can be characterised as “lovers of certification.”

## 4. Discussion

Food products possess various types of attributes that influence consumer purchase decisions. These attributes can be categorised as search, experience, or credence attributes. Credence attributes, such as animal welfare, organic production, or food safety, are not easily observable by consumers without the appropriate information and labelling [33]. Estimating consumers’ preferences for these attributes requires the application of choice experiment methods (CEMs), which enable the assessment of the willingness to pay [34]. CEMs offer several advantages in understanding consumer preferences for food attributes. They allow researchers to explore consumer preferences for attributes that may not currently exist in the market or are not readily available [35]. Additionally, CEMs provide a means to simulate observed markets and capture the complexity of multidimensional choices that consumers face [36]. These methods contribute to a better understanding of consumer behaviour and aid in informing marketing strategies and product development.

The preliminary data on fresh fruit consumption have provided valuable insights into the fruit market, especially concerning kiwi. The survey revealed a high inclination to purchase and consume fresh fruit in general, including kiwi, which aligns with the existing literature [4]. Among the different types of kiwifruits, yellow-pulp kiwi is the most well known and purchased, followed by organic and red-pulp kiwi. Approximately one-third of the sample indicated that they purchase or consider purchasing all three types of kiwifruits. This propensity is positively associated with the frequency of kiwifruit purchase in general (including green-pulp kiwi) and the overall attitude towards fruit consumption. These findings suggest that marketing strategies can leverage consumers’ attitudes towards fruit and kiwi to stimulate the specific markets for yellow-pulp, organic, and red-pulp kiwi.

Regarding red-pulp kiwi, which is the focus of the survey, a considerable percentage of individuals who are already aware of red-pulp kiwi either purchase them or have the intention to buy them. What is equally intriguing is the significant percentage of respondents who expressed uncertainty about purchasing red-pulp kiwi. Targeted marketing actions can be directed towards these two groups. Firstly, efforts can be made to convert the intention to purchase into an actual purchase. Secondly, strategies can be implemented to change the current disposition of undecided individuals and encourage them to develop an intention or take the step to purchase red-pulp kiwi. One potential approach could involve enhancing the knowledge and awareness of this relatively lesser-known fruit [37].

The analysis of respondents’ socioeconomic variables provides valuable insights for planning effective marketing actions. Among these variables, income level appears to have the most significant influence on purchase preferences, particularly for red-pulp and organic kiwi. In some cases, the level of education also influences choices to some extent. However, when considering all other variables in the analysis, apart from income, no statistically significant differences were observed.

These findings suggest that income level plays a crucial role in shaping consumer preferences for red-pulp and organic kiwi. Therefore, marketing strategies should take into account the income distribution of the target market to tailor pricing, promotions, and communication efforts accordingly. While education level may have a more subtle impact, it should not be overlooked as it could still influence consumer attitudes and behaviours to some degree [38].

A specific section of the research focused on assessing the attitudes of respondents towards fruit, kiwi in general, and each of the three types of kiwis examined. These attitudes were measured through a series of proposed items, such as “I eat kiwi for its vitamin C content” and “I like to try new foods and flavours.” The participants were asked to indicate their level of agreement or disagreement with these statements.

The analysis of the collected data revealed that the purchase behaviour, whether actual or probable, of yellow-pulp, red-pulp, and organic kiwi is influenced by the overall attitude towards kiwi. This attitude, in turn, is explained by the attitude towards fruit in general.

In summary, the respondents’ general attitude towards fruit influences their attitude towards kiwi, which subsequently affects their likelihood of purchasing yellow-pulp, red-pulp, and organic kiwi [39].

Several socioeconomic variables were found to be significant in influencing purchase behaviour, and it is crucial to consider them when planning effective marketing actions [40]. Based on the findings, the following recommendations are made:To increase the consumption of yellow-pulp, red-pulp, and organic kiwi, it is important to enhance the attitude towards kiwi among the elderly population, particularly in Germany. This can be achieved by emphasizing the benefits of kiwi that may be perceived as “more difficult” or less commonly known, such as their effects on sleep quality, their laxative effect, and their versatility in various food preparations.For yellow-pulp kiwi, communication activities should be initiated or strengthened, especially in Italy, Germany, and France. These countries require targeted efforts to increase awareness and stimulate purchase intentions. However, it is worth noting that Spaniards show a higher propensity for purchasing yellow-pulp kiwi, suggesting that communication activities may be relatively less needed in this market.In the case of red-pulp kiwi, focused communication activities are necessary, particularly in Spain and France, with a specific target audience of women. Strategies should be designed to raise awareness and appeal to women’s preferences and interests.To promote the consumption of organic kiwi, targeted campaigns should be implemented in Spain and France, specifically aimed at the elderly population with lower levels of tertiary education and lower incomes. These campaigns should emphasise the benefits and value of organic kiwi and tailor messages to resonate with the needs and preferences of this specific demographic.

By considering these recommendations and tailoring marketing activities accordingly, it is possible to encourage an increased consumption of yellow-pulp, red-pulp, and organic kiwi among different target groups.

Moreover, it was found that being male, being under 25 years of age, having a lower level of education, low income, and being of German nationality have a negative impact on attitudes towards fruit. Since these attitudes influence the overall perception of kiwi, including yellow-pulp, red-pulp, and organic varieties, marketing strategies aimed at promoting the consumption of these types of kiwis should focus on improving the attitudes towards fruit among these specific groups. This can be achieved by highlighting the “more difficult” aspects of fruit, such as the versatility of using fruit in various food preparations and emphasising their nutritional values.

Using the statistical tool of choice experiments, this study explored the preferences of respondents for different “red kiwi product systems” that vary in terms of relevant attributes, including origin area, brand, production method, certification, and price. The results reveal that the brand of a local enterprise is the most important attribute in the choice of purchasing red-pulp kiwi, followed by organic cultivation and the presence of a quality or sustainability certification [41]. Furthermore, it is important to note that the respondents derived greater utility from choosing one of the proposed kiwis than from not making a choice (“non-choice”). This implies that the availability of different options for red-pulp kiwi positively influences consumer preferences and satisfaction.

Finally, the analysis using “latent class” modelling revealed the presence of three distinct consumer groups. These groups, in decreasing order of probability of membership, include “Confident in local brands.” This group consists of individuals who have a preference for local enterprise brands over international brands and are willing to pay a premium for this preference [42]. This group primarily consists of young individuals, women, and residents of our country. People in this group exhibit a preference for local brands, value organic production methods, and consider price as an important factor.

The second group, identified as “Lovers of certification”, prioritises quality or sustainability certifications of kiwi. This group is willing to pay a premium for this attribute and also appreciate organic kiwi. Price remains a significant consideration for this group as well.

Lastly, the “Supporters of local sustainability” group places importance on the presence of quality or environmental sustainability certifications. This group also values both the Friuli Venezia Giulia region of origin and organic cultivation. Price plays a significant role for this group, and it predominantly comprises women aged 24 or older.

## 5. Conclusions

Green-pulp kiwifruit is the most well known and commonly purchased type, followed by organic and yellow-pulp kiwi. The introduction of red-pulp kiwi is relatively recent, and a significant portion of the sample expressed interest in purchasing or considering all three types of kiwifruits. This propensity is positively associated with the frequency of kiwifruit purchases in general. Therefore, marketing strategies can leverage consumers’ attitudes towards kiwis to promote the specific market for yellow-pulp, organic, and red-pulp kiwi. Regarding red-pulp kiwi, a notable percentage of individuals who are already aware of it are purchasing or intending to purchase it. However, there are respondents who remain undecided about purchasing. Targeted marketing actions can be implemented to improve the marketing of red kiwi, considering the significant influence of various socio-economic variables on purchase behaviour.

## Figures and Tables

**Figure 1 foods-12-02865-f001:**
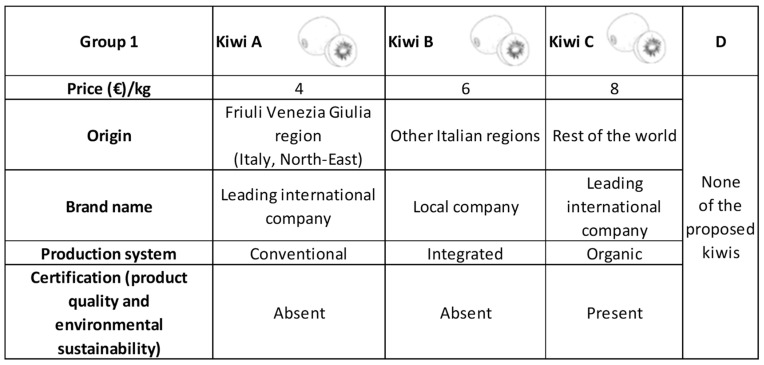
Examples of choices available for the consumers.

**Table 1 foods-12-02865-t001:** Attributes and levels of red kiwifruits used in the choice experiment.

Attributes	Attributes’ Levels
Price/kg	EUR 4.00
	EUR 6.00
	EUR 8.00
Area of origin	Friuli Venezia Giulia
	Other Italian regions
	Foreign countries
Brand	Leading international company
	Local business
Production method	Conventional
	Organic
Certification (product quality, environmental sustainability)	Presence
	Absence

**Table 2 foods-12-02865-t002:** Questionnaire used in this study submitted to the participants.

Do You Buy FRESH FRUITS?	Yes/No		
If yes;	only for yourself	only for other	both
How often do you eat FRESH FRUIT (one answer only)?			
several times a day			
regularly, as a snack			
regularly, usually at meals			
occasionally			
never			
Do you buy KIWIFRUITS?	yes/no		
If yes;	only for yourself	only for other	both
If yes, where do you prefer to buy KIWIFRUITS (one answer only)?			
at the farmer/grower			
in specialized shops			
at the supermarket/discount store			
at the local market			
other			
How often do you eat KIWIFRUITS (one answer only)?			
several times a day			
regularly, as a snack			
regularly, usually at meals			
occasionally			
never			
In which season do you usually eat KIWIFRUITS (one answer only)?			
all year round			
preferably in the summer			
preferably in winter			
other			
Do you know organic KIWIFRUITS?	yes/no		
If yes, do you buy them?	yes/no		
Are you familiar with yellow KIWIFRUITS?	yes/no		
If yes, do you buy them?	yes/no		
Do you know red KIWIFRUITS?	yes/no		
If yes, do you buy them?	yes/no		

**Table 3 foods-12-02865-t003:** Multinomial logit model and latent class model estimations; CLASS 1: young sustainability supporters, CLASS 2: confidence in local brands, CLASS 3: certification enthusiasts. ***, **, and * denote significance, respectively, at a 99%, 95%, and 90% confidence level.

	*MNL*	Latent Class Model					
		Young Sustainability Supporters		Confidence in Local Brands		Certification Enthusiasts	
Variable	Coeff. (S.E.)	Coeff. (S.E.)	WTP (EUR/kilo)	Coeff. (S.E.)	WTP (EUR/kilo)	Coeff. (S.E.)	WTP (EUR/kilo)
ASC	*−1.956 (0.049) ****	0.459 (0.370)	/	−2.521 (0.133) ***	/	−7.800 (1.217) ***	/
Price	*−0.307 (0.007) ****	−0.503 (0.060) ***	/	−0.115 (0.009) ***	/	−0.924 (0.050) ***	/
Friuli V.G.	*0.085 (0.007) ***	0.488 (0.196) **	0.98	0.028 (0.042)	/	0.027 (0.189)	/
Local enterprise	*0.372 (0.048) ****	0.279 (0.313)	/	0.609 (0.056) ***	5.30	0.050 (0.275)	/
Certification	*0.179 (0.039) ****	0.553 (0.244) **	1.10	0.058 (0.052)	/	1.129 (0.322) ***	1.22
Organic	*0.319 (0.038) ****	0.455 (0.241) *	0.91	0.395 (0.044) ***	3.44	0.389 (0.231) *	0.42
Average probability		0.19		0.45		0.36	
Theta in class probability model:							
Female		0.501 (0.173) ***		0.311 (0.141) **		0.00 (fixed parameter)	
Age 25–34		0.355 (0.296)		1.005 (0.216) ***		0.00 (fixed parameter)	
Age 18–24		−1.066 (0.491) **		0.854 (0.246) ***		0.00 (fixed parameter)	
Live in Italy		0.203 (0.206)		0.419 (0.170) **		0.00 (fixed parameter)	
Fruit consumption—daily		−0.307 (0.174) *		−0.092 (0.144)		0.00 (fixed parameter)	

## Data Availability

The data used to support the findings of this study can be made available by the corresponding author upon request.

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
