# Peer review of "Assessing Consumer Preferences for New Red-Pulp Kiwifruit: Application of a Choice Experiment between Different Countries"

_foods, 2023, doi:10.3390/foods12152865_

Round 1
Reviewer 1 Report (Previous Reviewer 3)
Dear Authors,
The manuscript (food-2530450) submitted for eview is quite interesting, and could be published after revision in the Foods Journal. However, it should be mentioned in the text that the characteristics of the respondents participating in the research were presented in the Supplementary material. The presented conclusions from the research are very enigmatic and very general. In my opinion it should be improved.
Not all references are cited in accordance with the journal's rules.
Despite my comments, I can recommend the manuscript for publication after minor revision.
Reviewer
Author Response
dear reviewer, attached the file with our responses. Many thanks, all the best.

Reviewer 2 Report (Previous Reviewer 1)
All my remarks were addressed.
Author Response
Many thanks for your kind work. All the best
Reviewer 3 Report (New Reviewer)
The Authors have chosen the topic of red-pulp kiwifruit, to better understand the consumers’ choices in a cross-cultural study.
The introduction gives a clear overview about the status of the kiwi growing sector and the main fruit types, mentioning the red-pulp variety.
Lines 68-69: Reference 17 is very outdated, please replace with a more recent one.
Lines 197-199: To present the different types of red kiwis, three identical, basic, and colourless designs were used. – It would be probably more effective, if the kiwi fruits on the displayed questionnaires would be pictures of red-pulp samples. I understand that in this experiment all choice sets were only red fruits, so the fruit’s flesh colour was not an experimental factor. However it is well known that the product’s colour has a strong effect on consumer acceptance.
Technical issues:
Lines 48-50 have yellow highlight, please remove it.
Lines 141-148 have yellow highlight, please remove it.
Line 265 have yellow highlight, please remove it.
Table 3 heading and footer have yellow highlight, please remove it.
Conclusion have yellow highlight, please remove it.
Ethical statement have yellow highlight, please remove it.
Author Response
dear reviewer, attached the file with our responses. Many thanks, all the best.

Reviewer 4 Report (New Reviewer)
The authors have presented a very detailed methodology which is easy to follow and read, one comment from my side is to provide subheaders so that it is easier for the readers to know which one is the experimental design, or which one is the statistics section.
L209 How did the author collect the CEM data? Online? Details needed
Can the authors provide also the rest of the choices as appendix/supplementary file on top of the examples in L202? Just to be clear to the readers what options are available for the participants.
Can the authors elaborate on how LCM can be used if only each participant only selected 1 option from the CEM experiment? How did the author carry or lay-out the dataset for analysis?
Table 3. Typo - sustaianability to sustainability
The classes needs be shuffled up before Table 3 to avoid confusion. I wasn't sure which class is what until I read in P10.
Author Response
dear reviewer, attached the file with our responses. Many thanks, all the best.

This manuscript is a resubmission of an earlier submission. The following is a list of the peer review reports and author responses from that submission.
Round 1
Reviewer 1 Report
The manuscript tries to investigate consumers' attitudes toward coloured kiwi fruits in a multinational European context. The research topic is well validated and interesting, however, several shortcomings should be addressed.
- Among the central objectives of the paper it is stated to "extend understanding of how developed and developing economy market consumers' values for credence attributes compare". However, investigations are made only in four developed countries (Italy, Spain, France and Germany)
- Figure 1 is supposed to illustrate a choice set, however, the "no choice" option is missing
- Given the nature of data collection, I suggest o use 'respondent' instead of 'interviewee'
- It is not clear why only one single kiwi production region of Italy is selected (Friuli-Venezia Giulia) when other European kiwi producer countries are also part of the survey
- Many of the tables rather should belong to the Appendix (Table 2, 4-9)
- The result of the MNL model should be named instead of using technical names ( Class1,2 and 3)
- It is unclear why Italian sample is introduced and discussed more (e.g., Table4 - regional distribution only available for Italy)
- There are several misspellings and inappropriate use of terms is also detected (e.g., in line 30, World Food and Agricultural Organizaton, instead of Food and Agriculture Organization of the United Nations)
- A separate section of Conclusion is required to clearly list the policy and managerial implications
Author Response
Dear anonymous reviewer. You can find our responses in the attached text. Many thanks for your suggestions, we tried to do our best to accomplish every point raised. King regards

Reviewer 2 Report
This paper is interesting, but it lacks academic rigor and some important parts. This makes the results less reliable. Here are some specific areas for improvement:
Line 27: There is a punctuation error in "fruit. where". It should be "fruit, where".
Line 36: The phrase "Italy is the second producer" should be modified to "Italy is the second largest producer".
Lines 66-96: This section of literature review is somewhat confusing and outdated. There are few references from the last five years, and many relevant studies on fruit attribute preferences based on choice experiments are not mentioned.
Lines 97-103: This section should be moved to "Materials and Methods".
The attributes and levels used in the choice experiment should be introduced clearly, such as how the price levels were set. In addition, the meaning of certification is not clear. It may be difficult for the respondents to understand this attribute, which reduces the credibility of the paper.
The price levels in Table 1 are 4, 8, and 10. However, there is a price of 6 in Figure 1. The same inconsistency applies to the integrated attribute.
The research method should be explained in more detail, such as what kind of econometric model was used and how it was estimated.
The paper should explain what the stars mean in Table 3. Stars usually indicate the significance level of the coefficients.
Author Response

(The authors gave the same response as above.)

Reviewer 3 Report
Dear Authors,
The manuscript (foods-2474217) subbmited for review is a quiet of interesting. Although I have mixed fealings to recommend it for publication. Perhaps the manuscript will be better after the Authors' answers to the questions and after major corrections.
Abstract
I don’t understand this sentence in the abstract: „The central objective of this paper is to extend the understanding of how developed and developing economy market consumers' values for credence attributes compare, and their preferences for red-pulp kiwifruit”. What did the authors mean they wrote developed and developing economy market. The research in the article only points developed markets.
Introduction:
The Introduction should indicate gaps in the research of other authors.
Results
The characteristics of the respondents are not a results, but the methodology of research, why the authors have included these data in the Results section.
In the results section, an analysis of consumer behavior depending on gender, age, level of education, monthly income class are needed. What is the point of asking respondents in detail, e.g. about the type of employment, if then this relationship is not taken into account.
From the manuscript, I didn't find out what the differences are between countries.
Conclusion
Conclusion should be a separate section.
References: References are cited according to journal rules.
Despite my comments, I am pleased to recommend this manuscript for publication but after a major revision. I believe it addresses an important area of research in an international context.
Reviewer
Author Response
Dear reviewer, please find our responses in the attached file. Many thanks, King regards

Round 2
Reviewer 1 Report
Most of my comments were addressed; however, there are some minor remarks remaining:
- in the MNL model, classes still have technical names (Class 1, 2 and 3). I would suggest to create descriptive names
- the conclusion section is not well developed and some unclear and not-supported statements might be confusing (e.g., yellow kiwifruit is the most known - in the sample, in general I suppose it is the green), etc.
Reviewer 3 Report
Dear Authors,
The authors changed many parts of the planned paper according to my suggestions.
I would like to thank the authors for considering my comments and applaud them for the major revisions to improve their manuscript. In my opinion, the manuscript is now clear and more understandable than the previous version.
I can recommend the manuscript for publication without correction.
Reviewer